

# Surprise Floods: The role of our imagination in preparing for disasters

Joy Ommer[1,2], Jess Neumann[1], Milan Kalas[2], Sophie Blackburn[1], Hannah L. Cloke[1,3]

[1]Department of Geography and Environmental Science, University of Reading, Reading, RG6 6UR, United Kingdom
[2]KAJO s.r.o., Bytca, 01401, Slovakia
[3]Department of Meteorology, University of Reading, Reading, RG6 6UR, United Kingdom

*Correspondence to*: Joy Ommer (j.ommer@pgr.reading.ac.uk)

**Abstract.** What's the worst that could happen? After a flood has devasted communities, those people affected, the news media, and the authorities often say that what happened was beyond our imagination. Imagination encompasses the picturing of a situation in our mind linked with the emotions that we connect with this situation. However, the role imagination actually plays in disasters remains unclear. In this regard, we analysed the responses of a survey which was disseminated in the 2021 flood affected areas of Germany. Some respondents perceived that due to their lack of imagination of the flood, they did not take adequate action in advance. This indicates that imagination plays an important role in disaster preparedness. Limited or lacking imagination could be linked to never having experienced a flood before, difficulties in interpreting forecasts and warnings, the perceived distance to waterbodies, and cognitive biases such as wishful thinking. Based on these results, we recommend future research should investigate to what extend visual support can advance forecast and warning communication in triggering the imagination of citizens in short-term. In a long-term perspective, research should focus how to cultivate imagination over time through participatory risk management, developing climate storylines, citizen weather observations, and the like.

## 1 Introduction

Devastating floods around the world are often reported to be "beyond our imagination" (The News International, 2022; United Nations, 2023; Dhakal, 2023; ClimateChangePost, 2021). In science communication and storytelling studies, this expression of beyond imagination is primarily used to highlight disasters for which the scale and the impacts are unknown, unexpected, or a complete surprise (Kundzewicz et al., 1999; Hollnagel and Fujita, 2013; Merz et al., 2015; de Bruijn et al., 2022; Cologna et al., 2017).

Despite the common use of the term imagination and a vast amount of literature in disciplines such as psychology, philosophy, and arts, the concept of imagination is not explored in depth in disaster research. Yet, our imagination plays an important part in preparing for uncertain futures through picturing threats and possible adaptations or disaster preparedness actions (Heino et al., 2022). Imagination usually refers to our ability to visualise a situation in our mind (Finn et al., 2023). Besides picturing a situation and possible actions, imagination is closely linked to our senses and how we might feel in this situation taking these



actions (Nanay, 2016). This ability to travel through time, picture, and test various scenarios strengthens us in anticipating and planning our future (Taylor, 2011).

We are living in a world where the future can turn into uncountable possible scenarios, and this makes us feel uncertain about our actual future (Yusoff and Gabrys, 2011). Forecasts and warnings of severe weather aim to support us in grasping likely
future scenarios, and there is an assumption that imagining these scenarios will make us take preparative actions. However, even if forecasts and warnings are received by citizens, (and sometimes they are not), they may not trigger the imagination of the impacts of the severe weather, and this means that people may not prepare for them. An example of this are the floods in Germany in July 2021, when devastating deadly floods occurred in western Europe due to a stagnating low pressure causing heavy rainfall of up to 180mm in 72h (Junghänel et al., 2021; Kreienkamp et al., 2021). The intense precipitation and resulting
flooding were both forecasted in advance for Germany at the national as well as the European level (Thieken et al., 2023). Yet, the flooding took thousands of people by surprise because many of them, foremost, did not receive any warnings, or perhaps more importantly, did not take the forecast or warning seriously or could not understand or imagine the consequences of the forecasted flooding (Cloke, 2022; Fekete and Sandholz, 2021).

*‚Es war klar, dass viel Regen kommt. Mir fehlte die Vorstellungskraft, was das bedeutet.‘*
        *(In English: 'It was clear that a lot of rain was coming. I lacked the imagination of what that means.')*
        *(Survey participant from Bad Neuenahr-Ahrweiler)*

If forecasts and warnings are not always effective and do not always steer people to be able to imagine and prepare for serious
floods (de Bruijn et al., 2022; Thieken et al., 2023), then we need to understand why. To address this research gap, this study aims to explore the role imagination plays in preparing for floods based on the responses of a semi structured online survey disseminated in areas affected by the 2021 flooding in Germany. In particular, this study seeks to identify barriers to the imagination of a hazard and how this affects the preparedness of citizens. Additionally, the study aims to distil possibilities for improving the communication of severe weather through forecasts and warnings to trigger imagination and to bridge the gap
to early action.

First, we frame the concept of imagination in Section 2. Then, we present the case study, the online survey, and its analysis in Section 3, and the results in Section 4. The main outcomes of the study are concluded in Section 5.

## 2 Imagination

What is imagination? In the context of this study, it can be described as the ability to depict a particular situation in your mind
and your actions linked to that situation (Nanay, 2016). For example, depicting river floodwater rushing into your basement and you are consequently evacuating yourself and your family to safety upstairs. Imagination also encompasses the emotions that this depiction of a flood might raise in us (Nanay, 2016), like the worries about the valuable things being flooded in your



basement or the fear of no knowing how high the water will rise and whether you and your family will be safe on the second floor. You yourself might have just been imagining this flood as you read this paragraph.

Creating these kinds of images in our mind is a cognitive capability and process that we are commonly applying and referring to as imagination (Finn et al., 2023). We use our imagination in our daily life, especially, in decision making. We tend to select the options which have a positive outcome, are not costly, are within our (perceived) capabilities, and that might even have additional benefits for us (Sunderrajan and Albarracín, 2021; Wang et al., 2021; Kuhlicke et al., 2020; Heidenreich et al., 2020). This way of decision making exemplifies a more controlled or rational behaviour compared to a decision made in panic

(Sunderrajan and Albarracín, 2021).

We draw on imagination voluntarily to try to depict how an episode of the future might look like (de Vito and Della Sala, 2011). You may not imagine several days of flooding and everything that might happen during those days but rather a moment such as sitting on your roof, crying, and waiting for help. Yet, imagining exactly this episode might be building on your previous experiences which pop up as mental imagery in your mind (Nanay, 2021). Our imagination may draw on previous

flooding experiences (if there are any) but is not confined to them (Finn et al., 2023). Thus, mental imagery can support us in creating images of potential futures in our mind (Cavedon-Taylor, 2021).

## 2.1 What shapes our imagination?

The way we imagine is not only shaped by our abilities of imagination but also by external and internal influences. Commonly, we develop our abilities to imagine from early childhood (Taylor, 2011). While every person may have different abilities,

extreme forms of imagination exist and some people have a very vivid imagination, which is known as hyperphantasia, while others may not have any imagination at all (aphantasia) (Palermo et al., 2022).

External influences can shape our imagination which are increasingly explored in research on imaginaries. For instance, geographical imaginaries explain that our imagination is shaped by spatial aspects i.e., how we think and feel about a place (Walshe et al., 2023). This concept can be further bridged to the controverse discussions around the influence of the proximity

to a risk area on risk perception (O'Neill et al., 2016; Ali et al., 2022; Rana et al., 2020). For instance, whether people living next to a river have a higher risk perception than people living far away from it.

Our imagination can be directed by personal factors. For instance, for some people, the trauma caused by past flood experiences can restrict their ability to picture the future in their minds (Gotlib, 2021). While for other people, the experience of previous floods can cause future threats to repeatedly reappear in their imaginations, resulting in hypervigilance (Mehring et al., 2023).

Imagination as a cognitive ability can also be hampered by wishful thinking, the attribution of reality to what one wishes to be true, even though it is not likely, for instance, when we think nothing bad will happen to me because floods aren't things that are likely to happen, and everything will be alright. Imagination can also be restricted by the availability bias, for example when we draw on our recent flood experiences and assume all future floods will be exactly like those (Merz et al., 2015). In reality, different floods can be very different experiences indeed. We usually overestimate the risk of potential future flooding



if we have experience of previous floods, while we underestimate the risk if we have no experience (Fischhoff et al., 1982;
Nanay, 2016).

## 2.2 Imagination & risk perception

Imagination is rarely discussed directly in disaster research. However, risk perception is a closely linked concept, which refers
to our belief about the potential risk from a flood (de Guttry and Ratter, 2022; Bulley and Schacter, 2021). At first glance,
imagination and risk perception may seem interchangeable, but in fact imagination plays a part in our (flood) risk perception
(Bulley and Schacter, 2021). It is acknowledged that risk perception is primarily influenced by reality and our factual
knowledge, such as locations of areas of flood risk, while imagination takes risk perception much further by adding the mental
picturing of a flood and the emotional component (the feelings that may be triggered by this mental picturing) (Karlsson et al.,
2023; Sobkow et al., 2016).

Risk perception may be lower, if the imaginative part is not triggered, for instance, if listening to or watching weather forecasts
does not result in a depiction of the hazardous impacts. Although some weather forecasts and warnings now explicitly try to
communicate impact (Potter et al., 2018a; Speight et al., 2021), this is far from universal and most weather forecasts and
warnings around the world still present information in a meteorological fact driven way, for example 40mm rain in an hour,
or a rise in the river of 1m in 1 day (WMO, 2015). This is despite the WMO calling for the global implementation of impact-
based forecasting and warning (WMO, 2015). The difficulties in translating what might seem like an arbitrary amount of
rainfall into a mental picture (and potential emotions), may lead us to perceive a lower risk. As we have seen, this translation
could be affected by a lack of knowledge or experience but also by cognitive biases or obstacles such as trauma. However, in
some cases past flooding experiences can benefit both sides of risk perception – the factual and the imaginational – through
the gained knowledge and mental imagery, respectively.

Risk perception is a prominent factor used to explain individual actions and motivation for preparing for flooding (Felletti and
Paglieri, 2019; Bubeck et al., 2013). Although risk perception is not the solely factor for taking preparedness actions, it can
lead to inaction if flood risk is perceived to be low (Kox et al., 2015). Nonetheless, even if we perceive that there is a risk of a
severe flood, it does not automatically trigger us to act. For instance, we might perceive the flooding to be so severe that we
believe our capabilities are not enough to take any or sufficient action, i.e. action is pointless because the outcome will be the
same; disastrous (Kuhlicke et al., 2020).

## 2.3 Triggering and cultivating imagination

Considering that our imagination can influence our flood preparedness behaviour, how exactly might this occur? Using photos
of previous floods is known to be one effective strategy for communicating warnings, especially, if these photos are from areas
close by to those people receiving the warnings (Kuller et al., 2021). As we have seen, impact-based forecasting aims to depict
the potential impact of an approaching flood and the implementation of such an approach was strongly recommended after the
Germany 2021 floods (Apel et al., 2022). Seeing the potential extent of the floods or impact on maps or similar ways of





visualisation may help us in creating mental images of potential flooding and may increase the uptake of disaster preparedness actions. This digital visual support is further explored with tools such as virtual and augmented reality, or digital twins (Bakhtiari et al., 2024; Mol et al., 2022; Skinner, 2020).

As we have seen, imagination is known to develop over time throughout our childhood and daily life, therefore, it is more commonly researched from a long-term perspective (Dobraszczyk, 2017; Finn et al., 2023; Taylor, 2011; Higueras and Molina Villaverde, 2022). In particular, disaster imagination can be cultivated through longer term interactions with people and by drawing on approaches from the arts such as storytelling, narratives, or simulations which can be used for understanding problems (i.e., flood risk areas) and identifying solutions for those (Fleming et al., 2016; Lloyd Williams et al., 2017; Bø and

Wolff, 2020). An example for this is the adoption of storytelling in the climate storyline approach which is building on the unfolding of previous disasters or potential future (Shepherd et al., 2018).

Throughout this section, we conceptualise imagination as the capability of creating mental pictures of situations and potential actions while also attempting to feel what we would feel if the situation was reality. Our imagination can be supported by past

experiences visually stored in our memory, but it can also be influenced by different factors. This section has highlighted the close relationship between imagination and our risk perception and the question of whether imagination can be triggered by receiving weather forecasts and warnings to increase preparedness motivation. The triggering of imagination could also be supported with visualisations such as photos or videos, and can also be cultivated over time, for instance, through storytelling approaches.

**3 Methods**

**3.1 Case study: July 2021 flooding in Germany**

In July 2021, severe rainfall stagnated over western Europe (Germany, Belgium, Netherlands, France, Luxembourg) for several days. This followed a longer wet episode in the summer. In Germany, the two states Rhineland Palatinate (RP) and North Rhine-Westphalia (NRW) were primarily affected with up to 182mm of rainfall recorded in 72h (Junghänel et al., 2021). Due

to the saturated soils, the water could barely infiltrate (Kreienkamp et al., 2021). Especially, in the hilly regions, surface runoff led to flooding, landslides, and other hazards (Lemnitzer et al., 2021; Dietze et al., 2022; Ibebuchi, 2022). Different types of flooding occurred throughout the states: flash flooding in smaller hilly catchments, fluvial flooding of rivers and streams, pluvial flooding partly forming gullies and new streams (Dietze et al., 2022; Thieken et al., 2023).

The event turned into a devastating disaster. In total, it was estimated that $162km^2$ were flooded of which 35.6% were built-up

areas (He et al., 2022). The (flash) flooding took many people by surprise while more than 180 people lost their life and more than 760 were injured throughout RP and NRW (Lehmkuhl et al., 2022; Thieken et al., 2023).

The communication of forecasts and the dissemination of warnings was one major issue leading to the high impact of the disaster. The heavy rainfall and likely flooding extent were forecasted in advance through the European Flood Awareness



System (EFAS) and German Weather Service (Deutscher Wetterdienst) (Thieken et al., 2023). However, the trickling down
of the information from the forecasts to those who needed it on the ground, hit many obstacles such as power outages and the
lack of emergency sirens (Kuehne et al., 2021); missing information such as behaviour recommendations and misinformation
(Fekete and Sandholz, 2021); or the underestimation of the severity of the flooding by authorities and the public (Thieken et
al., 2023).

### 3.2 Online survey

To gain a better understanding of the perspective of citizens affected by the floods, an online survey was designed. The online
survey allowed the collection of responses over a large area. The survey was primarily designed for flood affected citizens of
18 years and older who lived in North-Rhine Westphalia and Rhineland Palatinate during the time of the flooding. These two
federal states were selected because they were most severely impacted by the floods in Germany. The survey was developed
in both German and English and approved by the ethical committee of the University of Reading (14[th] February 2022).
Following the approval, it was disseminated via social media channels (Facebook, Twitter, LinkedIn, and WhatsApp) between
March and July 2022 – less than one year after the event. The authors were aware of potential biases i.e., the age structure of
respondents due to the chosen social media dissemination strategy.

The survey (available in the Supplementary Material) included mainly open questions in order to give the affected citizens a
voice. Closed questions were only used in cases such as the collection of basic information, or when information was clearly
definable like the source of flooding. The questions were addressing the following topics: the flooding source, risk awareness,
preparedness, response, early warning dissemination and content, issues that arose and solutions for the future, perception of
roles and responsibilities, and basic questions (age, living situation, and postcode).

### 3.3 Data analysis

After preprocessing the data (translation, post code correction), the responses were analysed through descriptive statistics and
thematic analysis. Descriptive statistics were used to gain a quantitative understanding of actions. The thematic analysis (Braun
and Clarke, 2006) was applied to gain a deeper insight into the responses but primarily to distil overarching themes that arose
throughout several questions, especially, throughout the open questions. The thematic analysis aims to work across multiple
questions instead of analysing the responses of one question in isolation. This method was chosen to identify patterns and
important themes that citizens have pointed out within their responses. The analysis includes four steps: 1) familiarisation with
the collected responses; 2) initial coding in NVivo (release 1.7.1) and Microsoft Excel; 3) identification of themes, 4) the
distilling of overarching themes such as imagination in this case. The overarching theme of imagination emerged from coding
responses in NVivo while the subthemes discussed in Section 4 were identified by manually coding imagination related
responses in Microsoft Excel.



## 3.4 Responses

The survey received 438 responses of which four were filled in English and 434 in German. The majority (87.7%) of respondents have been living in NRW and 12.3% in RP. The respondents covered all age groups (18 years and above) that were invited to contribute (Fig. 1a). 65% of the participants were aged between 25 and 54 years which is slightly overrepresenting this age group compared to German demographics (Population in Germany, 2024). Even though, the survey was in an online format, it did not prevent older age groups (65+) from contributing (9%). About two thirds of the participants

owned a house during the time of the event while 22% rented an apartment (Fig. 1b). Fewer people were living in a rented house, owning a flat or living at their parents' home.

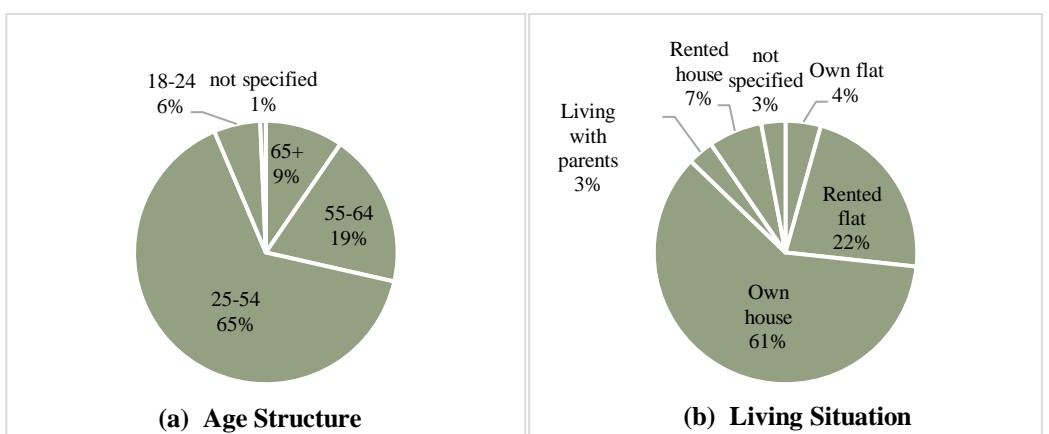

**Figure 1: (a) Age structure at the time of the survey; (b) living situation at the time of the event**


Almost all (96%) survey participants experienced flooding either directly or indirectly (e.g., through family, neighbours, and friends). The flooding was rated as extreme by 75% of the participants. More than half stated that they were directly affected by the flooding, and 250 people ticked that (also) their family, friends, or neighbours were affected. The businesses of 44 participants were flooded, and 262 respondents indicated that their daily life was affected by the flooding. Overall, three

quarters of the respondents selected from predefined options that they experienced extreme flooding, 19% declared that the flooding was worse than usual, 3% were affected by small or usual flooding, and 1% did not experience any flooding.

## 4 Results & discussion

The theme of imagination appeared in a number of different ways in the survey responses, revealing challenges in imagining extreme flooding, and allowing us to explore the connection between imagination and risk perception as well as disaster

preparedness, and finally, highlighting enablers and barriers to imagination.



### 4.1 Imagining a severe hazard

Some survey respondents realised that they were not able to imagine the flooding in advance because it was so severe.

*'Ich hatte es nicht so registriert, bis ich es sah…'*

*(In English: 'I hadn't understood it that way until I saw it…') (Euskirchen)*

The severity of the hazard was often linked with the limitations of imagining the hazard as it turned out to be which reflects the commonly used phrase beyond imagination. In particular, it was mentioned that the extent or dimension of the flooding was unimaginable.

*'Das Ausmaß könnte sich niemand vorstellen.'*

*(In English: 'No one could imagine the extent.') (Bonn)*

More specifically, the severity of the flood was not imaginable because the characteristics of the hazard, such as the depth, speed, and power of the water had not been previously experienced. For example, many people have not had the previous experience of walking through flood water.

*'[…] weil ich definitiv keine Vorstellung davon hatte, wie gewaltig Wasser sein kann.'*

*(In English: '[…] because I definitely had no imagination of how powerful water can be.') (Odenthal)*

Overall, the 'unknown' emerged as a prominent factor in people's experience of the flood, and this points to the limitations of our imagination, especially, in the context of previous flooding experiences. The unknown (the never experienced or expected) is what is often describe in news as 'beyond our imagination' (The News International, 2022; United Nations, 2023; Dhakal, 2023; ClimateChangePost, 2021; WDR Doku, 2022) which is also referred to as a surprise once it occurs (Merz et al., 2015). Hence, something unknown challenges our capabilities to imagine.

*'Die Wassermassen kannten wir nicht und waren bis dahin unvorstellbar'*

*(In English: 'We were not familiar with the masses of water, and until then, they were unimaginable.') (Aachen)*

Interestingly, even previous experiences of floods can limit our imagination, as survey participants showed that they could not imagine anything greater than what they are used to. This finding could be related to the claim that our imagination is limited through routines (Higueras and Molina Villaverde, 2022); thus, if a certain level of flooding is experienced a few times then

imagining it could be more severe is very difficult.

*'Weil Überschwemmungen hier in der Vergangenheit nicht so schlimm waren*

*und ich nicht damit gerechnet habe, dass das Wasser diesmal bedeutend höher steigt.'*

*(In English: 'Because flooding here hasn't been that bad in the past and I didn't expect the water*

*to rise significantly higher this time.') (Aachen)*



## 4.2 Imagination & risk perception


In many responses, it is difficult to distinguish between imagination and risk perception but, in the following statement, the person clearly expressed the fact that the personal underestimation of risk was also influenced due to the unimaginability of the flood (de Guttry and Ratter, 2022; Bulley and Schacter, 2021).

> *'Das Ausmaß der Katastrophe bis zum Schluss unterschätzt - es war im wahrsten (!) Sinne des Wortes*
> *UNGLAUBLICH und UNVORSTELLBAR!'*
> *(In English: 'The extent of the catastrophe underestimated until the end - it was literally (!) UNBELIEVABLE and*
> *UNIMAGINABLE!') (Bad Neuenahr-Ahrweiler)*

### 4.2.1 Place

Several respondents could not believe that they would be affected by the flooding, and this indicates that they perceived that
there was no risk. In many of these cases, this was because of the location of their home. For instance, it was far away from any flowing water or was even on a slope; thus, the respondents did not expect to be flooded. This proximity or distance to a risk area are commonly known as an influencing factor for risk perception, but the way it influences is not agreed on as studies show varying results (O'Neill et al., 2016; Ali et al., 2022; Rana et al., 2020). Our results show that the distance to water and living on the slope was often linked to lower perceived risk.

> *'Ich dachte nicht, dass es uns erreichen könnte, da der Bach eigentlich weit weg ist.'*
> *(In English: 'I didn't think it could reach us as the stream is actually far away.') (Weilerwist)*

This lower perceived risk due to distance was related to past experiences where the flood did not reach their home. Thus, they did not expect to be affected now. Here past experiences probably influenced the belief about these places, and this connects to the concept of geographical imaginaries in which we have a certain idea or perspective about the places around our home
which has evolved over time (Walshe et al., 2023).

### 4.2.2 Previous flood experiences

As we have seen, previous flooding experiences is known to influence risk perception, and people cannot imagine anything greater than they have seen before. Expanding upon this finding and it shows that by drawing on their experience, a false assessment of risk was estimated by respondents.

> *'Die Reaktionszeit war gleich Null, da wir in unserer Gegend nicht mit einer solchen Flutwelle gerechnet hatten.*
> *Beim Hochwasser 2016 waren wir überhaupt nicht betroffen.'*
> *(In English: 'The reaction time was zero because we did not expect such a flood wave in our area.*
> *We were not affected at all during the flood in 2016.') (Bad Neuenahr-Ahrweiler)*

Here, people are using their most recent experiences, in this example, this was the flooding of 2016 which was announced as
one of the most severe flooding of the Ahr River (Piper et al., 2016). Using past experiences in this way and gaining some





knowledge on flood behaviour can therefore also turn into a cognitive bias, an availability bias, limiting the imagination of a potentially more severe event (Merz et al., 2015). This further relates to the mental imagery which helped to imagine the flooding as it was in 2016 but nothing beyond.

### 4.2.3 Wishful thinking

Another cognitive bias that arose from the responses is wishful thinking. As we have seen, wishful thinking describes a cognitive bias in the belief that nothing significant will happen even though that a person may even expect that flooding will actually happen (Merz et al., 2015). We find that respondents could not believe that something significant would happen, and held onto the belief that all would be fine.

*'Ich konnte es wie so viele nicht glauben. Ich habe mir die ganze Zeit gesagt*

*es hört jetzt auf zu regnen und die Ahr geht wieder zurück.'*

*(In English: 'Like so many people, I couldn't believe it. I kept telling myself it would stop raining and*

*the Ahr would go back again.') (Bad Neuenahr-Ahrweiler)*

Interestingly, this quote perhaps implies that the person actually imagined what could happen and, therefore, has the hope that it won't happen and is deliberately blinding themselves to the risk. Additionally, this person shows an emotional aspect,

particularly of fear, which is likely to have increased the wishful thinking. However, more investigation would be needed to understand to what extent and in what ways this person actually imagined what could happen.

### 4.2.4 Flood mitigation measures

Another interesting finding which can be linked to previous flooding and risk perception is expressed in the following quote:

*'Unser Haus ist auf einem Sockel gebaut, der die letzte Flut aus den 80er Jahren berücksichtigt hat.*

*Wir dachten, das würde reichen.'*

*(In English: Our house is built on a pedestal that took into account the last flood from the 1980s.*

*We thought that would be enough.') (Aachen)*

The respondent mentions that the house was built in a way that it would be flood resistant because it was elevated. Therefore, it would be safe if it flooded to a similar way to the flood in the 1980s. However, this knowledge and sense of security that the

house would be safe in case of a flooding may have limited their imagination that the flooding could be worse, and the water depth could be even greater. This is another example of where the flooding could be characterised as beyond imagination. This respondent may not have experienced the flooding in the 1980s first hand, but still has the knowledge about the potential water depth. This was possible to imagine for this person. Hence, it shows that imagination does not exclusively build on previous experiences and mental imagery.





### 4.3 Imagination and preparedness


Limited imagination of the approaching threat caused inaction. A few people still took actions, often because of their previous flooding experience. However, the people who prepared for the event, mainly focused on last minute emergency measures.

#### 4.3.1 Inaction

The difficulties of imagining the threat itself can potentially be linked to inaction. Several people who expressed that they
could not imagine or realise the extent of the threat, mentioned that they did not prepare.

*'Ich war auf diese Wassereinbrüche nicht vorbereitet, weil ich definitiv keine Vorstellung davon hatte [...]'*

*(In English: 'I was not prepared for these water intrusions because I definitely could not imagine it [...]') (Odenthal)*

*'Im Vorfeld nichts [getan]- habe das Ausmaß nicht realisiert.'*

*(In English: '[Did] nothing in advance - I didn't realise the extent of it.') (Bad Neuenahr-Ahrweiler)*

*'Keiner war vorbereitet! Bzw. hat das Ausmaß nicht realisiert'*

*(In English: 'No one was prepared! Or rather, did not realise the extent of it.') (Bad Münstereifel)*

The term 'realise' implies the idea of making something real which can be closely linked to picturing the threat. The following quote highlights that received rainfall forecasts could probably not be imagined because the person was lacking knowledge or experience to translate this factual information into mental images.

*'Die angegebenen [Regen] Mengen pro Quadratmeter waren nicht richtig zu begreifen oder zu fassen.*

*Ich hatte keinerlei spezielle Vorkehrungen getroffen. '*

*(In English: 'The stated quantities [of rainfall] per square meter could not be understood or grasped correctly.*

*I hadn't taken any special actions.') (Euskirchen)*

Some responses showed that people may imagine the threat but cannot imagine any actions they could take because the threat
seems much greater than their own capabilities. This links directly to behavioural theory. People are motivated to protect themselves and their families based on both the personal threat that they perceive and their appraisal of their own capabilities to take action, their belief in what they are able to actually do (Kuhlicke et al., 2020). In the following quotes, the belief of being powerless was described, and this could express that people do not believe that their capabilities are sufficient, or the flood is perceived to be too severe.

*'Da kann man leider nichts tun, Man ist machtlos. [...] Man handelt irrational.'*

*(In English: 'Unfortunately there's nothing you can do, you're powerless. [...] You act irrationally.') (Zülpich)*

*'Wenn ein wirklich großes Hochwasser ansteht, ist man gegen die Kraft dessen machtlos,*

*egal wie gut man sich als individuelle Privatperson vorbereitet.'*

*(In English: 'If a really big flood is imminent, you are powerless against the force of it,*

*no matter how well you prepare as an individual.') (Kall)*



After experiencing this severe flooding, some people still cannot imagine any actions that they would be capable of taking to be prepared in the future.

*'[…] weil man sich da auch in Zukunft nicht drauf vorbereiten kann. Außer wegziehen.'*

*(In English: '[…] because you can't prepare for it in the future either. Except move away.') (Landkreis Vulkaneifel)*

One respondent mentioned that, especially, after this severe flooding, it would be impossible to imagine actions in case of an even worse flood.

*'Sobald jedoch mehr Infrastruktur beschädigt worden wäre, ist es immer noch schwer vorstellbar, was wir tun sollten.'*

*(In English: 'However, once more infrastructure had been damaged, it is still difficult to imagine what we should do.')*

*(Dahlem)*

Not knowing or imagining potential actions in preparedness or response led to irrational actions; thus, the ability to imagine possible worst-cases and actions that could be performed is important and therefore, needs to be well communicated, planned, and trained.

*'Klare Vorgaben für alle, es muss die Überlegung geben, dass so etwas passieren kann,*

*dieses Ereigniss war so nicht vorstellbar und war auch nie trainiert worden.'*

*(In English: 'Clear guidelines for everyone, there must be consideration that something like this can happen,*

*this event was unimaginable and had never been trained.') (Zülpich)*

### 4.3.2 Action

In contrast to the above, some respondents actually took actions despite the fact that they mentioned they could not imagine the threat. These actions were primarily emergency measures, and this may imply that the respondents at some point realised

the approaching flood.

*'Meiner Familie geholfen […]. Sandsäcke befüllt, Unterlagen gesichert.'*

*(In English: 'Helped my family […]. Sandbags filled; documents secured.') (Bad Neuenahr-Ahrweiler)*

*'Pumpen im Keller installiert; Autos in einer höher gelegenen Region geparkt.'*

*(In English: 'Pumps installed in the basement; cars parked in a higher area.') (Aachen)*

*'Sandsäcke befüllt und vor das Haus gelegt.'*

*(In English: 'Sandbags filled and placed in front of the house.') (Landkreis Ahrweiler)*

*'Außenanlagen gesichert.' (In English: 'Outdoor facilities secured.') (Euskirchen)*

Another reason that people prepared despite not being able to imagine the hazard extent can be explained with the availability bias. These people have experienced flooding once or several times before and were familiar with it; thus, they prepared

routinely.

*'Maßnahmen im Kellerbereich sind routiniert'*

*(In English: 'Measures in the basement area are routine') (Grafschaft)*



*'Die von vorherigen Starkregen-Ereignissen bekannten Schwachstellen gesichert.*

*War leider nicht ausreichend, da die Regenmenge zu viel war.'*

*(In English: 'The vulnerabilities known from previous heavy rain events have been secured. Unfortunately, it wasn't enough*

*because the amount of rain was too much.') (Aachen)*

*'Ich habe schon oft Hochwasser in diesem Haus erlebt, so dass ich eine gewisse*

*Routine und Gelassenheit bewahren konnte. […] So extrem kannte ich das dann doch noch nicht.'*

*(In English: 'I have experienced flooding in this house many times, so I have been able to maintain*

*a certain routine and composure. […] but this extreme was unknown to me.') (Sudern)*

This routine of preparing for floods interestingly demonstrated a rational and calm behaviour; they knew what they had to do. We have seen that previous experience limits the imagination of something more severe than the usual flooding, and here this shows the same effect but going one step further, that the people prepared as they usually do but since they could not imagine something more severe, they also did not prepare for a more severe event. They stayed in their familiar preparedness routine.

This was on the one hand very useful, but on the other hand the routine became a trap that limited imagination. Routines are known to be the enemy of imagination as they restrict thinking and imagination beyond the usual habits (Higueras and Molina Villaverde, 2022).

### 4.4 Imagining

The previous section has highlighted the importance of imagination for taking preparedness actions. Here, we explore to what

extent weather forecasts and warnings triggered imagination, and the understanding of local risks.

### 4.4.1 Triggering imagination through weather forecast and warning (short-term)

The forecasts and warnings about heavy rainfall and potential flooding were not always understood in the way expected by forecasters. Some respondents stated that hearing about the amount of projected rainfall did not trigger their imagination of what was about to happen.

*'Ich wusste das es viel regnen soll, konnte mir bei der Liter Angabe aber nicht drunter vorstellen, dass es SO viel sein*

*würde…' (In English: 'I knew it was going to rain a lot, but given the liters I couldn't imagine that it would be THAT*

*much...') (Erftstadt)*

Hence, hearing a certain number or a warning of the color purple was mentioned to be too abstract or vague to create an image in one's mind i.e., picturing how this number will change the water level. However, it remains unknown whether a number of

the water level would actually be useful for triggering imagination considering that the number of the forecasted rainfall amount was claimed as too abstract.

*'[…] die genannten Regenmengen von "bis zu 100l/m²" sind zu abstrakt […].'*

*(In English: '[…] the mentioned rainfall amounts of "up to 100l/m²" are too abstract […].') (Aachen)*



*'Die Markierung auf der Wetterkarte war tief lila. Sagt aber nichts über die Höhe des evtl. Wasserstandes aus.'*

*(In English: 'The marker on the weather map was deep purple. But it says nothing about the height of the possible water level.') (Bad Neuenahr-Ahrweiler)*

Imagining a situation can be easier if people are able to draw on their mental imagery, for instance, if people have experienced flooding before. Survey participants reported that receiving photos or videos of the flooding from friends or family helped them to picture what is happening, and this potentially helped them to imagine what may be about to happen in their own
localities.

*'[...] bewusst wurde es erst durch die Bilder aus Hagen.'*

*(In English: '[...] I only became aware of it through the pictures from Hagen.') (Euskirchen)*

*'20:45 Video von Altenahr erhalten und von dann das Wasser nicht aus den Augen gelassen.'*

*(In English: '20:45 video received from Altenahr and from then on I didn't take my eyes off the water.') (Dernau)*

In this example, the video was from an upstream location and only about 7.5km away. Hence, through watching the video, it was clear that this situation is real and is very likely to happen soon in the respondent's village. The spatial proximity of a source of information is known to be an effective way to trigger an alerting effect in people's minds (Kuller et al., 2021). Additionally, if the photo or video presents a situation which is familiar to a person, it can trigger the emotional aspect of imagination:

*'Ich erhielt ein kleines Video von einem Parkplatz, der unter Wasser stand.*
*Dort setzte sich ein Auto in Bewegung, was mich schockierte, da ich mir das Entsetzen des Besitzers vorstellte.'*

*(In English: 'I received a short video of a parking lot that was under water.*
*A car started moving there, which shocked me as I imagined the owner's horror.') (Bad Münstereifel)*

Illustrating the potential impact seems to be an important element in triggering our imagination of the potential threats:

*'Mehr darüber berichten und ggf. mal veranschaulichen, was es bedeutet, wenn 200l/qm runter kommen.'*

*(In English: 'Report more about it and if necessary, illustrate what it means when 200l/sqm comes down') (Erftstadt)*

As we have seen, a starting point for integrating visuals can be impact-based forecasting (Potter et al., 2018) and using virtual or augmented reality (Bakhtiari et al., 2024; Mol et al., 2022).

### 4.4.2 Cultivating imagination (long-term)

Working with visuals may be an effective way to enable us to imagine the threat of flooding, but this may not be enough. As we have seen some people can draw on their previous experiences (at least to some limited extent) which others do not have. The results discussed so far suggest that people need access to some factual knowledge and imagination to increase risk perception. Hence, a first step is to encourage people to learn more about rainfall amounts, flood levels, and how these relate to what happens in their own neighbourhood.

*'Weil ich mich mit den persönlichen Konsequenzen bis heute nicht konsequent auseinander gesetzt habe.'*

*(In English: 'Because I haven't consistently dealt with the personal consequences to this day.') (Bad Neuenahr-Ahrweiler)*



It may also be important for people to be more attentive to their own environment and observe the rain falling locally and how wet the landscape is. For instance, one person who experienced the flood now has developed their own rainfall threshold at which preparedness actions will be taken.

*'Ich würde anhand der zu erwartenden Regenmenge entscheiden. Bei den Mengen des letzten Jahres würde ich vorab schon*

*die Taschen sicherheitshalber packen und mein Umfeld warnen. Bei den üblichen Mengen (ca. 40l/m²) bleibe ich gelassen.'*

*(In English: 'I would decide based on the expected amount of rain. With the quantities of last year, I would already pack my*

*bags as a precaution and warn my surroundings in advance. With the usual amounts (about 40l/m²), I remain calm.')*

*(Euskirchen)*

Although not everyone has experienced severe rainfall and flooding; through their own regular observations people can gain a better understanding of what a specific rainfall amount communicated in forecasts and warnings can mean in someone's area or in upstream areas. In addition, people living close to a river or stream could start observing water levels and by comparing the forecasted levels with how the river looks in reality, they may gain a further understanding of what water level forecasts mean in reality.

*'Prognosen zu Überschwemmungsgebieten und Pegelständen sind wichtig.'*

*(In English: 'Forecasts of flood zones and water levels are important.') (Euskirchen)*

To raise awareness of risks or the need for environmental awareness and observation in a community, approaches such as storytelling could be used to identify local risks, unfold past hazards, or identify potential solutions to minimise risk. This could be combined within a participatory development of local climate storylines (Shepherd et al., 2018). This way,

imagination could be cultivated over time.

The quotes in this subsection on cultivating imagination could apply to everyone, although logically younger people may benefit most as they may have less experiences with extreme weather.

*'... gerade junge Leute können sowas ja nicht einschätzen was normal ist und was nicht, da viele bestimmt*

*nicht studieren wann wieviel Liter Regen runter kommt um dann so eine hohe Liter Angabe einschätzen zu können.'*

*(In English: '... young people in particular cannot assess what is normal and what is not, as many certainly do not study*

*when and how many liters of rain come down in order to be able to estimate such a high liter figure.') (Erfstadt)*

### 4.5 Limitations of the study

This exploratory study provided insights into the role of imagination in disaster preparedness by analysing a semi structured survey. Since the survey was primarily designed to gain an insight into early warning, preparedness, and response and the topic

of imagination only emerged from this survey, the analysis faced the following limitations which we recommend are considered for future research:

- The results of the survey sometimes provided some limited evidence on which speculative interpretations were necessary, and thus, those themes without fully comprehensive evidence should be explored in more depth in future





studies. For instance, the linkage influence of hazard knowledge on imagination or whether imagination of a hazard
can lead to wishful thinking.

- Some survey respondents expressed their emotions directly in their responses which could be partly linked to imagination. Emotions are a primary part of imagination (Nanay, 2016); thus, future studies should explore this in more detail. In this context, it is recommended to use other qualitative methods such as focus groups (Finn et al., 2023) or interviews (Walshe et al., 2023).

- Linkages to the idea of place and especially the proximity to hazard areas were found. Future research should focus on the external influence which different kind of imaginaries (social, political, historical, climate change) have on the imagination of specific disasters as discussed in this paper.

- Another recommendation is to further investigate the relationship between forecast uncertainty and imagination.

## 5 Conclusion

The primary ambition of this paper was to explore the role of imagination in disaster preparedness, as the term imagination is commonly used by media but not specifically researched in the context of disaster events. For this purpose, the paper builds on a survey that was disseminated in flood affected areas in Germany of 2021. In this paper, imagination is defined as our ability to picture a scenario and potential actions in our mind as well as the emotional consequence of them. The survey results indicate the difficulties that people had in imagining a severe flood and the consequences of this is that they did not take

preparedness actions. People's ability to imagine a severe hazard was mainly hampered because of an element of unknowing. In other words, survey participants showed difficulties imagining something they have not experienced before such as the power and speed of flood water, or the dimensions flooding can have. While previous experiences were found to be beneficial for imagination, it was also found to be a bias for some people, as respondents could not imagine something worse than what they have experienced so far, it was literally beyond imagination.

We find that imagination is closely linked to the concept of risk perception. The risk we perceive builds on our factual knowledge (gained through education or experience) and our imaginations. If we are not able to imagine a severe hazard, then most likely our risk perception will be lower. Our results suggest that our factual knowledge is often needed as a base or input for imagination. For instance, when hearing specific rainfall forecasts, it may not trigger our imagination if we cannot build on our factual knowledge which provides us with an understanding of what 200mm of rainfall in 1 day means.

Other barriers to imagining a flood were identified. Firstly, the spatial distance to a river or the location of a house on a slope prevented respondents from imagining that the flood would reach their home. Secondly, some respondents demonstrated a specific idea and belief about a place in which flooding was considered impossible. This finding links to the concept of geographical imaginaries. Thirdly, cognitive biases showed barriers to imagination such as wishful thinking (and desperate hope). Respondents believed that flooding will not happen often against the evidence and even though it was sometimes

perceived as very likely. Another cognitive bias that was implied was the availability bias, which is closely linked to previous



experiences of flooding and probably constitutes the one of the main thresholds for risk in people's minds. Here, people could neither believe nor imagine that a flood could be worse than the one they had already experienced; thus, it is likely that they were trapped in their mental imagery of the past.

A key finding of this work is that the reason that people did not take preparedness actions is most likely because they could
not imagine the flooding in advance. People who have experienced flooding before may have prepared, but mostly only for the flooding extent which they have previously experienced because they did not imagine that the flood could be worse.

This study showed that the imagination of something unknown poses a great challenge to many people. Therefore, it is important that weather forecasts and warnings can trigger imagination, which can help in perceiving risk and taking preparedness actions. To trigger imagination in the short-term, more research is needed on the communication of a severe
weather forecast and warning using the support of visual elements such as photos and videos, but also digital tools like virtual and augmented reality. These can support efforts in implementing impact-based forecasting and increase understanding of the dimensions of an approaching flooding. Our results show that locality is important and photos of a person's hometown or somewhere close by will likely make imagination of the flood easier. Furthermore, showing familiar elements such as a car which might be floating away can increase the understanding and imagination of what might be happening.

Finally, it is important to cultivate our imagination over time by continuously increasing our factual knowledge on risk. This can be supported by using creative approaches such as storytelling. For instance, local climate storylines could be co-developed with communities by discussing local risks, past flooding events, and potential flood mitigation options.

In conclusion, this study explored the role of imagination in disaster preparedness, highlighting that the imagination of unknown severe weather can pose difficulties and, therefore, constrain disaster preparedness. To gain a deeper understanding
of the barriers and enablers for imagination and how it can be incorporated in weather forecast and warning communication, more interdisciplinary research is needed. Research on imagination has the potential to transform the way in which forecasts and warnings are received, understood, and acted upon. If we can harness our power of imagination to help us prepare better for disasters, then we can save lives in future disasters.

**Data availability**

The participants of this study did not give written consent for their data to be shared publicly, so due to the sensitive nature of the research, the survey data is not available.

**Author contributions**

Joy Ommer (conceptualisation, investigation, visualisation, writing, review & editing); Jess Neumann (conceptualisation, supervision, review & editing); Milan Kalas (conceptualisation, supervision, review & editing); Sophie Blackburn
(conceptualisation, supervision); Hannah L. Cloke (conceptualisation, supervision, review & editing).





**Competing interests**

The authors declare that they have no conflict of interest.

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
