# Peer review of "Surprise Floods: The role of our imagination in preparing for disasters"

_EGUsphere, 2024_

## Referee Comment (RC2)

**NHESS Review**

**Surprise Floods: The role of our imagination in preparing for disasters**

**General comments**

Thank you for the opportunity to review this paper and congratulations on this work. It is a novel topic on the role of imagination that can help advance disaster risk communication and preparedness research to support policy and practice. The paper provides interesting results from the survey disseminated in the 2021 floods in Germany, through which the theme of imagination emerged. The paper demonstrates evidence around links between 1) imagination and risk perception and explores how to 2) cultivate/trigger imagination for risk communication (short-term and long term).

I really welcome the evidence shared in the paper but it could be improved by clarifying the aims of the paper and how these flow with the literature sections, results/discussion sections and conclusions. For example barriers are mentioned as an aim but not framed as a results/ discussion section and then return in conclusions. Risk perception is framed in the result/ discussion and literature section(s) but not mentioned as an aim, and is described in the conclusions. Improving communication is mentioned in aims, literature and conclusions but not in the results/discussion.

I feel the paper has a lot of potential to propose a conceptual framework (incl. a figure) for linking imagination and risk perception or to use an existing theory e.g. protection motivation theory and enhance it by reflecting on the role of imagination. I leave this up to the authors, but the paper would benefit from a stronger theoretical framing especially because this is such a novel/ emerging topic.

**Specific comments**

Introduction

From line 38 the German case is discussed which is also discussed in 3.1. I suggest the authors reconsider adding these details in the introduction and remove the quote as the results should not already be displayed here. If there are examples or learning around imagination and preparedness for disasters from other countries this would be more suitable. You could also consider moving up the imagination section 2 first para to this intro section.

As mentioned above, I suggest the aims of the paper are written more clearly to reflect the evidence drawn from the study and this thread is clearer through the paper in subheadings etc.

A clear connection to risk perception in the introduction would be helpful to prepare the reader for it coming up strongly later in the paper.

*Section 2*

Section 2.2. Suggest delving deeper to the theories/ models around risk perception and protective action for disasters e.g. Protection motivation theory (Bubeck et al. 2017; Bubeck et al. 2012), protective action decision model (Lindell and Perry, 2012). The paper only mentions 'behavioural theory' (line 320) without due reflection on this in the literature.

Section 2.3. Suggest exploring the risk communication literature – for example Balog-Way (2020) can signpost you to different approaches and theories and Kellens et al. (2013) providing a review of

perception and communication of flood risk. Your text is talking about using cultivating imagination through risk communication but not explicitly saying it, so I suggest you consider this framing.

Additional references to arts & disasters literature could be added e.g. Sevilla et al (2023) on envisioning.

*Section 3 Methods*

I suggest mentioning in 3.2 that the survey/ questions were not designed specifically around imagination, as this was a theme that that emerged from the data. This ties in with the limitations of the study, where you frame it as an 'exploratory study'. I suggest this reflection on limitations is moved up to methods and focus the final section on recommendations or implications for future research.

Section 3.4 – I don't think the information on living situation is needed – or has it emerged as important in the findings linked to imagination? I also suggest removing the figure which doesn't add value to the paper. I would like to see more information about the percentage of responses that mentioned imagination and if possible any other interesting characteristics of this group specifically.

In relation to limitations and more linked to results/discussion, it would be helpful to know what percentage of responses are informing the different themes e.g. in 4.2.1. it says 'several' respondents and in others just provide one quote, which makes it harder to understand how represented the sub-themes were in the data. You do mention in limitations that some themes did not have 'fully comprehensive evidence' Line 468 so it important to be upfront about this when discussing any such theme in the results/ discussion.

Suggest indicating your process for referring to the participants in the documentation (quotes) to ensure anonymity.

I'm not clear on how descriptive statistics was used to understand actions and if these results are presented in the paper – please clarify.

*Section 4 Results and Discussion*

Overall I feel that the results and discussion subsections would benefit from more reflection and connection back to the literature.

In Section 4.1/4.2/4/3 I'd like to see some restructuring here to delve deeper into the key messages on imagination emerging from the data - rather than all of them – and focus in on risk perception – which to me seems like the first key aim/ and theoretical contribution of the paper. For example, Section 4.1 could strengthen 4.2.2 around the link with between risk perception and flood experience, which comes up again in 4.3 the action/inaction section.  Putting all this together can make a stronger argument about the link between imagination and previous experience – and then how this links to risk perception. In connection with this and as mentioned in the methods section, it would be helpful to know the percentage of respondents that did not have previous experience of flooding before this event and then to understand how this affected their ability to imagine the event – put perhaps this is out of scope. I would also suggest cutting some quotes and focus in on the most interesting ones, and discuss in connection to the literature.  As mentioned previously, it

would be very interesting to create a conceptual figure/framework for linking risk perception and imagination.

Line 301 – 'Limited imagination of the approaching threat caused inaction.' Suggest changing caused to influenced as many other factors (out of the control of citizens) mentioned earlier also influenced the lack of action e.g. lack of flood warnings.

Section 4.4 for me this section focuses on the second aim of the paper to explore how we can trigger imagination via risk and warning communication. I suggest a stronger sub-title for 4.4 and a clearer intro to the section to reflect the aim. It would be helpful to remind the reader about the forecasts that were/not available and I suggest the authors draw more on risk communication literature as mentioned in the lit section and also literature specifically around communicating early warnings e.g. Parker et al. 2009. Again, here I suggest reflecting on whether all the quotes are needed and to expand out the discussion for example, on using visuals/ mental imagery and storytelling within risk communication practices to trigger imagination in the short and long-term. Some reflection on the policy and practical implications would be interesting to add here too.

*Section 4.5 Limitations*

As mentioned above I suggest a section on implications for research or something along those lines. I recommend not using bullet points to state the different research areas. Suggest to focus in on future research around risk perception and risk communication to enhance preparedness for disasters.

**Technical corrections**

- Abstract line 17 change extend to 'extent'
- Line 63 – 'not' knowing
- Line 84 – 'controversial'
- Line 276 – remove 'that' in even though that

Please check for any additional typos/ grammatical errors.

Some areas of the paper read more like a report or thesis rather than a research article, so I suggest the authors consider this when revising.

*References for consideration:*

Balog-Way, D., McComas, K., & Besley, J. (2020). The evolving field of risk communication. Risk Analysis, 40(S1), 2240-2262.

Bubeck, P., Wouter Botzen, W. J., Laudan, J., Aerts, J. C., & Thieken, A. H. (2018). Insights into flood-coping appraisals of protection motivation theory: Empirical evidence from Germany and France. *Risk analysis*, *38*(6), 1239-1257.

Kellens, W., Terpstra, T., & De Maeyer, P. (2013). Perception and communication of flood risks: A systematic review of empirical research. *Risk Analysis: An International Journal*, *33*(1), 24-49.

M.K. Lindell, R.W. Perry, The Protective Action Decision Model: Theoretical Modifications and Additional Evidence, Soc. Risk Anal. 32 (2012) 616–632

P. Bubeck, W.J.W. Botzen, J.C.J.H. Aerts, A Review of Risk Perceptions and Other Factors that Influence Flood Mitigation Behavior, Risk Anal. 32 (2012) 1481–1495. doi:10.1111/j.1539-6924.2011.01783.x.

Parker, D. J., Priest, S. J., & Tapsell, S. M. (2009). Understanding and enhancing the public's behavioural response to flood warning information. *Meteorological Applications: A journal of forecasting, practical applications, training techniques and modelling*, *16*(1), 103-114.

Sevilla, E., Jarrín, M. J., Barragán, K., Jáuregui, P., Hillen, C. S., Dupeyron, A., ... & Sevilla, P. N. (2023). Envisioning the future by learning from the past: Arts and humanities in interdisciplinary tools for promoting a culture of risk. *International journal of disaster risk reduction*, *92*, 103712.

---

## Author Comment (AC2)

**Response to comments on egusphere-2024-296**

'Surprise floods: the role of our imagination in preparing for disasters'
by Joy Ommer, Jess Neumann, Milan Kalas, Sophie Blackburn, and Hannah L. Cloke,
EGUsphere, 2024

**Referee: Anonymous Referee #2**

**Referee comments and our responses**

**General comments**:

Thank you for the opportunity to review this paper and congratulations on this work. It is a novel topic on the role of imagination that can help advance disaster risk communication and preparedness research to support policy and practice. The paper provides interesting results from the survey disseminated in the 2021 floods in Germany, through which the theme of imagination emerged. The paper demonstrates evidence around links between 1) imagination and risk perception and explores how to 2) cultivate/trigger imagination for risk communication (short-term and long term).

*Response 1: Thank you for valuing our work!*

I really welcome the evidence shared in the paper but it could be improved by clarifying the aims of the paper and how these flow with the literature sections, results/discussion sections and conclusions. For example barriers are mentioned as an aim but not framed as a results/ discussion section and then return in conclusions. Risk perception is framed in the result/ discussion and literature section(s) but not mentioned as an aim, and is described in the conclusions. Improving communication is mentioned in aims, literature and conclusions but not in the results/discussion.

*Response 2: Thank you for pointing this out! We will make the aim, results, and conclusions more coherent.*

I feel the paper has a lot of potential to propose a conceptual framework (incl. a figure) for linking imagination and risk perception or to use an existing theory e.g. protection motivation theory and enhance it by reflecting on the role of imagination. I leave this up to the authors, but the paper would benefit from a stronger theoretical framing especially because this is such a novel/ emerging topic.

*Response 3: Thank you for proposing these two options! We will strengthen the linkage between risk perception and imagination (probably also adding a figure) and add a reference to behavioral theories such as the PMT.*

**Specific comments:**

**Introduction**

From line 38 the German case is discussed which is also discussed in 3.1. I suggest the authors reconsider adding these details in the introduction and remove the quote as the results should not already be displayed here. If there are examples or learning around imagination and preparedness for disasters from other countries this would be more suitable. You could also consider moving up the imagination section 2 first para to this intro section.

As mentioned above, I suggest the aims of the paper are written more clearly to reflect the evidence drawn from the study and this thread is clearer through the paper in subheadings etc.

A clear connection to risk perception in the introduction would be helpful to prepare the reader for it coming up strongly later in the paper.

*Response 4: Since the paper builds on the German case study, we would not remove it from the introduction and neither add more details there as it might disrupt the flow of the introduction. We can add some of the few example studies/learnings on imagination from other areas/countries and introduce the concept of risk perception. In regard to the aim, please see Response 2.*

**Section 2**

Section 2.2. Suggest delving deeper to the theories/ models around risk perception and protective action for disasters e.g. Protection motivation theory (Bubeck et al. 2017; Bubeck et al. 2012), protective action decision model (Lindell and Perry, 2012). The paper only mentions 'behavioural theory' (line 320) without due reflection on this in the literature.

*Response 5: We will make a reference to these but will not go into detail since the results do not have sufficient evidence to completely wrap them around behavioral theories.*

Section 2.3. Suggest exploring the risk communication literature – for example Balog-Way (2020) can signpost you to different approaches and theories and Kellens et al. (2013) providing a review of perception and communication of flood risk. Your text is talking about using cultivating imagination through risk communication but not explicitly saying it, so I suggest you consider this framing.

Additional references to arts & disasters literature could be added e.g. Sevilla et al (2023) on envisioning.

*Response 6: Thank you for these suggestions, we will integrate them.*

**Section 3 Methods**

I suggest mentioning in 3.2 that the survey/ questions were not designed specifically around imagination, as this was a theme that that emerged from the data. This ties in with the limitations of the study, where you frame it as an 'exploratory study'. I suggest this

reflection on limitations is moved up to methods and focus the final section on recommendations or implications for future research.

*Response 7: Thank you for these suggestions, we will integrate them.*

Section 3.4 – I don't think the information on living situation is needed – or has it emerged as important in the findings linked to imagination? I also suggest removing the figure which doesn't add value to the paper. I would like to see more information about the percentage of responses that mentioned imagination and if possible any other interesting characteristics of this group specifically.

*Response 8: We will remove the figure/information suggested and aim to add more information on participants mentioning imagination, but it may cause difficulties extracting descriptive statistics from the qualitative data.*

In relation to limitations and more linked to results/discussion, it would be helpful to know what percentage of responses are informing the different themes e.g. in 4.2.1. it says 'several' respondents and in others just provide one quote, which makes it harder to understand how represented the sub-themes were in the data. You do mention in limitations that some themes did not have 'fully comprehensive evidence' Line 468 so it important to be upfront about this when discussing any such theme in the results/ discussion.

Suggest indicating your process for referring to the participants in the documentation (quotes) to ensure anonymity.

I'm not clear on how descriptive statistics was used to understand actions and if these results are presented in the paper – please clarify.

*Response 9: The results presented in Section 4 are all coming from the qualitative data (open questions). Only the results presented in Section 3.4 were derived from closed questions and therefore, descriptive statistics could be applied. We will try to be more specific about the number of participants informing about different themes. Not having fully comprehensive evidence was referring to the fact that more contextual information (from the participant(s)) would have been needed to make an evidenced link to existing theories.*

**Section 4 Results and Discussion**

Overall I feel that the results and discussion subsections would benefit from more reflection and connection back to the literature.

In Section 4.1/4.2/4.3 I'd like to see some restructuring here to delve deeper into the key messages on imagination emerging from the data - rather than all of them – and focus in on risk perception – which to me seems like the first key aim/ and theoretical contribution of the paper. For example, Section 4.1 could strengthen 4.2.2 around the link with between risk perception and flood experience, which comes up again in 4.3 the action/inaction section. Putting all this together can make a stronger argument about the link between imagination and previous experience – and then how this links to risk

perception. In connection with this and as mentioned in the methods section, it would be helpful to know the percentage of respondents that did not have previous experience of flooding before this event and then to understand how this affected their ability to imagine the event – put perhaps this is out of scope. I would also suggest cutting some quotes and focus in on the most interesting ones, and discuss in connection to the literature. As mentioned previously, it would be very interesting to create a conceptual figure/framework for linking risk perception and imagination.

*Response 10: We appreciate the comments and aim to integrate them. However, we don't have statistics regarding previous flooding experiences as there was no question asking the participants about it. Some mentioned in questions that they had previous experience, but we cannot derive any percentages from this as it would be incomplete. We will add this to the limitations.*

Line 301 – 'Limited imagination of the approaching threat caused inaction.' Suggest changing caused to influenced as many other factors (out of the control of citizens) mentioned earlier also influenced the lack of action e.g. lack of flood warnings.

*Response 11: We will rephrase this sentence.*

Section 4.4 for me this section focuses on the second aim of the paper to explore how we can trigger imagination via risk and warning communication. I suggest a stronger sub-title for 4.4 and a clearer intro to the section to reflect the aim. It would be helpful to remind the reader about the forecasts that were/not available and I suggest the authors draw more on risk communication literature as mentioned in the lit section and also literature specifically around communicating early warnings e.g. Parker et al. 2009. Again, here I suggest reflecting on whether all the quotes are needed and to expand out the discussion for example, on using visuals/ mental imagery and storytelling within risk communication practices to trigger imagination in the short and long-term. Some reflection on the policy and practical implications would be interesting to add here too.

*Response 12: Thank you for these suggestions, we will aim to integrate them.*

**Section 4.5 Limitations**

As mentioned above I suggest a section on implications for research or something along those lines. I recommend not using bullet points to state the different research areas. Suggest to focus in on future research around risk perception and risk communication to enhance preparedness for disasters.

*Response 13: Thank you for these suggestions, we will adjust this section for implications for future research.*

**Technical corrections**

- - Abstract line 17 change extend to 'extent'
- - Line 63 – 'not' knowing

- • - Line 84 – 'controversial'
- • - Line 276 – remove 'that' in even though that

Please check for any additional typos/ grammatical errors.

Some areas of the paper read more like a report or thesis rather than a research article, so I suggest the authors consider this when revising.

*Response 14: Thank you for these suggestions, we will aim to integrate them.*

***Thank you for the valuable comments!***

---

## Author Response (AR2)

Dear Mr. Sakic Trogrlic,

Thank you for reviewing our article and accepting it for publication!

In regard to your minor revision request, we added a figure depicting the case study area. In addition, we included the number of survey responses received from the districts within the study area.

All the best,

Joy Ommer

---

## Author Response (AR3)

Dear Mr Sakic Trogrlic,

Thank you for your request!

We added the elements to the study area (Figure 1). Please, let us know in case it remains unclear.

All the best,

Joy